Hemogram data as a tool for decision-making in COVID-19 management: applications to resource scarcity scenarios

Avila Eduardo 1 2 3 e.avila@edu.pucrs.br
http://orcid.org/0000-0002-7556-7904 Kahmann Alessandro 3 4
Alho Clarice 1 3
http://orcid.org/0000-0001-8534-3480 Dorn Marcio 3 5
1 Forensic Genetics Laboratory, School of Health and Life Sciences, Pontifical Catholic University of Rio Grande do Sul , Porto Alegre, RS , Brazil
2 Technical Scientific Section, Federal Police Department in Rio Grande do Sul , Porto Alegre, Rio Grande do Sul , Brazil
3 National Institute of Science and Technology - Forensic Science , Porto Alegre, Rio Grande do Sul , Brazil
4 Institute of Mathematics, Statistics and Physics, Federal University of Rio Grande , Rio Grande, Rio Grande do Sul , Brazil
5 Laboratory of Structural Bioinformatics and Computational Biology, Institute of Informatics, Federal University of Rio Grande do Sul , Porto Alegre, Rio Grande do Sul , Brazil
Suner Aslı
Electronic publication date: 2020 Jun 29
Publication date: 2020
Volume: 8
Electronic Location ID: e9482
Received 2020 May 10; Accepted 2020 Jun 15
Copyright: © 2020 Avila et al.
Copyright year: 2020
Copyright holder: Avila et al.
License: This is an open access article distributed under the terms of the Creative Commons Attribution License, which permits unrestricted use, distribution, reproduction and adaptation in any medium and for any purpose provided that it is properly attributed. For attribution, the original author(s), title, publication source (PeerJ) and either DOI or URL of the article must be cited.
License URL: https://creativecommons.org/licenses/by/4.0/

Keywords: COVID-19, Machine learning, Naïve-Bayes, Hemogram, Scarcity

Funding: The authors received no funding for this work.

==============================
Background

COVID-19 pandemics has challenged emergency response systems worldwide, with widespread reports of essential services breakdown and collapse of health care structure. A critical element involves essential workforce management since current protocols recommend release from duty for symptomatic individuals, including essential personnel. Testing capacity is also problematic in several countries, where diagnosis demand outnumbers available local testing capacity.

Purpose

This work describes a machine learning model derived from hemogram exam data performed in symptomatic patients and how they can be used to predict qRT-PCR test results.

Methods

Hemogram exams data from 510 symptomatic patients (73 positives and 437 negatives) were used to model and predict qRT-PCR results through Naïve-Bayes algorithms. Different scarcity scenarios were simulated, including symptomatic essential workforce management and absence of diagnostic tests. Adjusts in assumed prior probabilities allow fine-tuning of the model, according to actual prediction context.

Results

Proposed models can predict COVID-19 qRT-PCR results in symptomatic individuals with high accuracy, sensitivity and specificity, yielding a 100% sensitivity and 22.6% specificity with a prior of 0.9999; 76.7% for both sensitivity and specificity with a prior of 0.2933; and 0% sensitivity and 100% specificity with a prior of 0.001. Regarding background scarcity context, resources allocation can be significantly improved when model-based patient selection is observed, compared to random choice.

Conclusions

Machine learning models can be derived from widely available, quick, and inexpensive exam data in order to predict qRT-PCR results used in COVID-19 diagnosis. These models can be used to assist strategic decision-making in resource scarcity scenarios, including personnel shortage, lack of medical resources, and testing insufficiency.

Introduction

Since its first detection and description (Huang et al., 2020), COVID-19 expansion has brought worldwide concerns to governmental agents, public and private institutions, and health care specialists. Declared as a pandemic, this disease has deeply impacted many aspects of life in affected communities. Relative lack of knowledge about the disease particularities has led to significant efforts devoted to alleviating its effects (Lipsitch, Swerdlow & Finelli, 2020).

Alternatives to mitigate the disease spread include social distancing (Anderson et al., 2020). Such a course of action has shown some success in limiting contagion rates (Tu et al., 2020). However, isolation policies manifest drawbacks as economic impact, with significant effects on macroeconomic indicators and unemployment rates (Nicola et al., 2020). To address this, governments worldwide have proposed guidelines to manage the essential workforce, considered pivotal for maintaining strategic services and provide an appropriate response to the pandemics expansion (Black et al., 2020).

Widespread reports of threats to critical national infrastructure have been presented, with significant impact associated with medical attention (Kandel et al., 2020). Significant pressure is being faced by emergency response workers, with some countries on the brink of collapse of their national health systems (Tanne et al., 2020). The main concern associated with COVID-19 is the lack of extensive testing capacity. Shortage of diagnostic material and other medical supplies pose as a major restraining factor in pandemics control (Ranney, Griffeth & Jha, 2020).

The most common COVID-19 symptoms are similar to other viral infectious diseases, making the prompt clinical diagnostic impractical (Adhikari et al., 2020). Official guidelines emphasize the use of quantitative real-time PCR (qRT-PCR) assays for detection of viral RNA in diagnosis as the primary reference standard (Tahamtan & Ardebili, 2020). In many countries, test results are hardly available within at least a week, forcing physicians and health care providers to take strategic decisions regarding patient care without quality information.

Previous reports have described alterations in laboratory findings in COVID-19 patients. Hematological effects include leukopenia, lymphocytopenia and thrombocytopenia, while biochemical results show variation on alanine and aspartate aminotransferases, creatine kinase and D-dimer levels, among other parameters (Guan et al., 2020; Huang et al., 2020). Some efforts have been applied to evaluate clinical and epidemiological aspects of this disease using computational methods, such as diagnosis, prognosis, symptoms severity, mortality, and response to different treatments. A useful review of some of these methods is presented by Wynants et al. (2020).

The main objective of this article is to provide insights to healthcare decision-makers facing scarcity situations, as a shortage of test capacity or limitations in the essential workforce. A useful method of doing so is using hemogram test results. This clinical exam is widely available, inexpensive, and fast, applying automation to maximize throughput. To do so, we have analyzed hemogram data from Brazilian symptomatic patients with available test results for COVID-19. We propose a framework using a Naïve-Bayes model for machine learning, where test conditions can be adjusted to respond to actual lack of resources problems. Finally, four distinct scarcity scenarios examples are presented, including handling of the essential workforce and shortage of testing and treatment resources.

Materials and Methods

Data collection

A total of 5,644 patients admitted to the emergency department of Hospital Israelita Albert Einstein (HIAE, São Paulo, Brazil) presenting COVID-19-like symptoms were tested via qRT-PCR. A total number of 599 patients (10.61%) presented positive results for COVID-19. The full dataset contains patients anonymized ID, age, qRT-PCR results, data on clinical evolution, and a total of 105 clinical tests. Not all data was available for all patients, therefore the amount of missing information is significant, with most available parameters informed for a small fraction of subjects. All variables were normalized (i.e., mean = 0 and variance = 1) to maintain anonymity and remove scale effects. No missing data imputation was performed during model generation to avoid bias. Considering the significant amount of missing data, only 510 patients presented values for all 15 parameters evaluated in hemogram results (comprising the following cell counts or hematological measures: hematocrit, hemoglobin, platelets, mean platelet volume, red blood cells, lymphocytes, leukocytes, basophils, eosinophils, monocytes, neutrophils, mean corpuscular volume (MCV), mean corpuscular hemoglobin (MCH), mean corpuscular hemoglobin concentration (MCHC), and red blood cell distribution width (RDW). Data for the above parameters were used in model construction, along with qRT-PCR COVID-19 test results. No baseline characteristics detailing can be presented for the evaluated cohort, once additional patients description is not accessible. Even the limited provided individual information (as subjects ages, for instance) was normalized within the dataset, and therefore cannot be inferred from available data. The full dataset is available in https://www.kaggle.com/einsteindata4u/covid19 and can also be found in Supplemental Materials.

Machine learning analysis: Naïve Bayes classifier

Machine learning (ML) is a field of study in computer science and statistics dedicated to the execution of computational tasks through algorithms that do not require explicit instructions but instead rely on learning patterns from data samples to automate inferences (Mitchell, 1997). These algorithms can infer input-output relationships without explicitly assuming a pre-determined model (Geron, 2017; Hastie, Tibshirani & Friedman, 2009). There are two learning paradigms: supervised and unsupervised. Supervised learning is a process in which the predictive models are constructed through a set of observations, each of those associated with a known outcome (label). In opposition, in unsupervised learning, one does not have access to the labels, it can be viewed as the task of “spontaneously” finding patterns and structures in the input data.

Our objective with this study is to predict in advance the results of the qRT-PCR test with a supervisedmachine learning model using data from hemogram tests performed on symptomatic patients. The main process can be divided into four steps: (1) pre-processing of the data (2) selection of an appropriate classification algorithm, (3) model development and validation, that is, the process of using the selected characteristics to separate the two groups of subjects (positive for COVID-19 vs. negative for COVID-19 in qRT-PCR test), and (4) test generated model with additional data. Steps are detailed as follows:

Data pre-processing

Samples presenting a missing value in any of the 15 evaluated features were removed, in order to avoid bias introduction in model. A total of 510 patients (73 positives for COVID-19 and 437 negatives) presented complete data and were considered for the model construction.

Classification algorithm

In this work, we use a Gaussian Naïve Bayes (NB) classifier, which is a probabilistic machine learning model used for classification tasks. The main reasons for choosing this classifier are due to its low computational cost, its ability to handle missing data and because it presented better classification performance when compared to other evaluated ML techniques, for this particular dataset (data not shown). In medicine, the first computer-learn attempts in decision support were based mainly on the Bayes theorem, in order to aggregate data information to physicians’ previous knowledge (Martin, Apostolakos & Roazen, 1960). The Naïve Bayes (NB) method combines the previous probability of an event (also called prior probability, or simply prior) with additional evidence (as, e.g., a set of clinical data from a patient) to calculate a combined, conditional probability that includes the prior probability given the extra information. The result is the posterior probability of an outcome, or simply posterior. This classifier is called “naïve” because it considers that each exam result (variables) is independent of each other. Since this situation is not realistic in medicine, the model should not be interpreted (Schurink et al., 2005). Besides this drawback, it can outperform more robust alternatives in classification tasks, and once it reflects the uncertainty involved in the diagnosis, Bayesian approaches are more suitable than deterministic techniques (Gorry & Barnett, 1968; Hastie, Tibshirani & Friedman, 2009).

Model development and validation

A classifier is an estimator with a predict method that takes an input array (test) and makes predictions for each sample in it. In supervised learning estimators (our case), this method returns the predicted labels or values computed from the estimated model (for this work, positive or negative for COVID-19). Cross-validation is a model evaluation method that allows one to evaluate an estimator on a given dataset reliably. It consists of iteratively fitting the estimator on a fraction of the data, called training set, and testing it on the left-out unseen data, called test set. Several strategies exist to partition the data. In this work, we used the Leave-one-out (LOO) cross-validation model, as in Chang et al. (2003), since this method is appropriate to handle small sample size datasets. The number of data points was split N times (number samples). The method was trained on all the data except for one point, and a prediction was made for that point. The proposed approach was implemented in Python v.3 (https://www.python.org) code using Scikit-Learn v. 0.22.2 (Pedregosa et al., 2011) as a backend.

Missing data tolerance

In order to evaluate the adequacy and generalization power of the proposed model, as well as its tolerance to handle samples containing missing data (i.e., at least one variable with no informed values), an additional set of 92 samples (10 positives for COVID-19 and 82 negatives) was obtained from the patient database. Those samples were not initially employed in model delineation, considering they present a single missing value among all 15 employed hemogram parameters. This “incomplete dataset”, comprising 92 samples with a single missing value information per sample, was then submitted to the previously generated model, in order to evaluate classification performance and missing values handling ability for the model.

Results

Descriptive analysis

For data description, probability density function (PDF) of all 15 hemogram parameters were estimated through the original sample by kernel density estimator. Some hemogram parameters present notable differences between the distributions of positive and negative results, mainly regarding its modal value (distribution peak value) and variance (distribution width). Differences are summarized in Table 1. Regarding basophiles, eosinophils, leukocytes and platelets counts, qRT-PCR positive group distribution shows lower modal value and lower variance. On the other hand, monocyte count displays opposite behavior, once lower modal value and variance are observed for the qRT-PCR positive group. The remaining nine hemogram parameters did not show a notable difference between negative and positive groups. All variables contribute to the classification model, and despite the fact classification can be perfomed without the complete set of parameters (i.e., including missing data), a most sucessfull prediction is achieved when complete hemogram information is used as input. PDF analysis results are presented in Fig. 1.

Table 1 Descriptive analysis of hemogram parameters used in present study.

Parameter	Modal value	Variance	
Basophiles	Reduced in positive cases	Reduced in positive cases	
Eosinophiles	Reduced in positive cases	Reduced in positive cases	
Leukocytes	Reduced in positive cases	Reduced in positive cases	
Monocytes	Augmented in positive cases	Augmented in positive cases	
Platelets	Reduced in positive cases	Reduced in positive cases	
Note:

Parameters not shown displayed no difference between negative and positive cases.

Figure 1 Probability density function (PDF) of all 15 hemogram parameters.

(A) Basophils; (B) Eosinophils; (C) Hematocrit; (D) Hemoglobin; (E) Leukocytes; (F) Lymphocytes; (G) MCH; (H) MCHC; (I) MCV; (J) MPV; (K) Monocytes; (L) Neutrophils; (M) Platelets; (N) RDW; (O) Red Blood Cells.

Naïve Bayes model results

A NB classifier based on training set hemogram data was developed. Under the model, the complete range of prior probabilities (from 0.0001 to 0.9999 by 0.0001 increments) was scrutinized, and posterior probability of each class was computed for different prior conditions. A posterior probability value of 0.5 was defined as the classification threshold in one of the positive or negative predicted groups. Resulting model showed a good predictive power of the qRT-PCR test result based on hemogram data. Figure 2 shows the accuracy, sensitivity, F1 score, and specificity curves derived from the model for different prior probabilities of each class (positive or negative for COVID-19). Reported prior probabilities refer to positive COVID-19 condition.

Figure 2 Performance metrics of proposed Naïve-Bayes model.

Prior probabilities are presented in reference to positive qRT-PCR prediction. Confusion matrices (left to right) are presented for 0.9999, 0.2933 and 0.0001 prior probabilities, respectively. Sensitivity = True Positive Ratio; Specificity = True Negative Ratio.

When setting the prior probability to the maximum defined value (0.9999), the NB classifier correctly diagnosed all PCR positive cases. On the other hand, such configuration improperly predicted 77.3% of negative PCR results as positive. Regarding the lower possible prior probability setting, it does not classify a single observation as positive. This result can be explained by the unbalanced number of observations for each class, tending to over classify samples as the class with more observation, that is, negative results. Such characteristics can also be noticed in the general accuracy, since smaller values for the prior used in the classifier tend to diagnose all observations as belonging to the dominant class (negative) and consequently raising the total of correctly classified samples. The break-even point is met when prior probability is set to 0.2933. Under this condition, all metrics are approximately 76.6%.

Regarding the model sensitivity, the rate of positive samples correctly classified is over 85% within 0.999 to 0.5276 range, with small decrease of it when the prior probability of positive result is diminished within this range. When prior is set to values under 0.1, the number of positive predicted samples decreases rapidly, yielding lower sensitivity. As for specificity, it presents linear growth as tested priors decrease. Ultimately, the accuracy results profile are similar to specificity, due to the negative patients dominance. Figure 3 shows additional results for the classification performance presented by the proposed method. Figure 3A displays the ROC (receiver operating characteristics) curve for the methods. Obtained measure of the area under the curve (AUROC) is equivalent to 0.84, suggesting excellent prediction performance for the model. Figures 3B–3D presents prediction results for the baseline model, using a dummy classifier for the most frequent class (negative qRT-PCR, Fig. 3B), and for stratified (Fig. 3C) and uniform (Fig. 3D) evaluations.

Figure 3 Classification performance evaluation.

Classification perfomance was evaluated through ROC curve (A) or against the baseline model using a dummy classifier for (B) the most frequent class (negative); (C) stratified data; (D) uniform data. AUC = Area Under the Curve; Sensitivity = True Positive Ratio; Specificity = True Negative Ratio.

As mentioned above, prior probability choice has a critical relevance in proposed model use. It is clear that, when extreme values of positive probability are applied (close to 0 or 1), specific classes (positive or negative qRT-PCR test results predictions) are favored, increasing its ability of correct detection. As an example, when a value of 0.9999 is set for prior probability of positive result is set, an increase in misclassification in negative class results is observed. At the same time, it is possible to properly identify samples where hemogram evidence strongly indicates a negative result, according to the model. This is based on the fact that evidence used in the model construction (in present case, hemogram data) must strongly support the reduction of posterior probability of disease to values under 0.5, therefore leading to a negative result. This logic can be applied to fine tune the prior probability used in the model, in order to improve correct classification of positive or negative groups prediction. Examples of how to use this feature is provided in the ‘‘Discussion’’ section. Samples including a single missing value (n = 92, including 10 qRT-PCR positives) were used to test the missing data tolerance presented by the proposed model. Figure 4 presents results obtained from the model application to this incomplete (lacking information for one variable per sample) dataset.

Figure 4 Classification performance for training (LOO) and missing value-containing datasets.

Results presented for the complete prior probability range. Results are presented as the percentage of correctly predicted qRT-PCR exams. Informed prior probability refers to positive outcome. TN, true negative; TP, true positive.

Discussion

Laboratory findings can provide vital information for pandemics surveillance and management (Lippi & Plebani, 2020). Hemogram data have been previously proposed as useful parameters in diagnosis and management of viral pandemics (Shimoni, Glick & Froom, 2013). In the present work, an analysis concerning hemogram data from symptomatic patients suspected of COVID-19 infection was executed. A machine learning model based on Naïve Bayes method is proposed in order to predict actual qRT-PCR from such patients. The presented model can be applied to different situations, aiming to assist medical practitioners and management staff in key decisions regarding this pandemic, especially in conditions of limited access to medical resources (Brown, Ravallion & Van de Walle, 2020). Figure 5 summarizes model construction and application. Predictions are not intended to be used as a diagnostic method since this technique was designed to anticipate qRT-PCR results only. As such, it is highly dependant on factors affecting qRT-PCR efficiency, and its prediction capability is dependent on the sensitivity, accuracy, and specificity of the original laboratory exam (Sethuraman, Jeremiah & Ryo, 2020).

Figure 5 NB Model construction (A) and application diagram (B).

Descriptive analysis of hemogram clinical findings shows differences in blood cell counts and other hematological parameters among COVID-19 positive and negative patient results. Differences are conspicuous among three measures (leukocytes, monocytes and platelets) and more discrete to additional two (basophiles and eosinophiles). It is possible that differences are also present across the complete data spectrum, even though they are not clearly visualized with PDF data. These results are in accordance to previous reports of changes in laboratory findings in COVID-19 infected patients, where conditions as leukopenia, lymphocytopenia and thrombopenia were reported (Fan et al., 2020; Ding et al., 2020). It is important to highlight that data analysis is not sufficient to characterize clinical hematological alterations in evaluated patients (when compared to demographic hematologic parameters data), once data was normalized for the evaluated sample set only. However, even within this particular quota of population (individuals presenting COVID-19-like symptoms), differences were found between individuals presenting negative or positive qRT-PCR COVID test results. The proposed NB-ML model can be helpful in accessing different levels of information from hemogram results, through inferring non-evident patterns and parameter relationships from this data. Also, our simulations suggest that the NB model has at least some degree of tolerance to missing data values, which can be advantageous when compared to other ML techniques.

Bayesian techniques are based on the choice of a prior probability of an event (in present case, positive result for qRT-PCR test). The method considers actual evidence (hemogram data) to result in a posterior probability of the outcome (prediction of a positive result). By changing the selected prior probability, we can derive an uncertainty analysis of the model to understand its distribution. Uncertainty can be then applied to adequately adapt the classifier to a particular ongoing context. This option allows the evaluation of different decision-making scenarios concerning diverse aspects of pandemics management. During a crisis situation, measures should be taken seeking to maximize benefits and achieve a fair resource allocation (Emanuel et al., 2020). To illustrate the model flexibility and how it can be used to help on this matter, a general framework of application is proposed, followed by a simulation of four scenarios where resource scarcity is assumed.

Application framework

The proposed NB model can be applied in two distinct situations. When clinical data is available for a particular patient, it is highly recommended that medical staff determine the prior probability on a case-by-case basis. When no clinical or medical data is available, or when decisions regarding resource management involving multiple symptomatic patients are necessary, the model can be used in multiple individuals simultaneously, aiming to identify those with higher probabilities of presenting positive qRT-PCR results.

Individual assessment

Individual risk management and personal evaluation is essential for COVID-19 response (Gasmi et al., 2020). Individuals presenting COVID-19 symptoms are medically evaluated where no COVID-19 test is available for appropriate diagnosis confirmation. Medical practitioners can determine a probability of disease based on anamnesis, symptoms, clinical exams, laboratory findings and other available data. This probability of infection, as determined by the physician or medical team, can be considered as the prior probability. Using hemogram data as input, and informing the prior probability of COVID-19 based on medical findings, the model will consider hemogram data to inform a posterior probability, which can be higher or lower than the original, and based on the hemogram alterations caused by the virus infection. It is important that hemogram data would not be included in original medical assessment and prior determination, in order to avoid bias and reduce model overfit.

Multiple patients evaluation

It can be used in situations where decisions are necessary for resource management including multiple individuals. Choice of a target group (positive or negative qRT-PCR result prediction) should be defined. The model can be applied to multiple individuals simultaneously, with the choice of prior probability carefully adjusted to result in a specific number of predicted individuals from the target group, according to the desired outcome. This method increases the correct selection of candidates belonging to the target group, when compared to random selection. Individual results can be ranked based on posterior probability of a positive or negative result, and results stratified according to convenience, as a way to evaluate a particular scenario of interest. When additional clinical data is available, or become available later, patients selected during bulk evaluation should be reassessed individually as proposed in the general framework, in order to reduce misclassifications.

Applications to scarcity scenarios

Examples of proposed model use are presented for some specific scarcity scenarios in Table 2. As can be seen, the model sensitivity can be adjusted by selecting prior probability employed, according to desired outcome or interest group. Prior selection should be carefully decided, based on current context or situation proposed, and must consider the classification group where higher accuracy is intended.

Table 2 Strategies for NB-ML model applications and symptomatic patient selection in scarcity conditions.

Hemogram test results are available for all symptomatic patients. Scenarios proposed for situations where test results are not available (no testing or waiting qRT-PCR test results). Prediction results were appraised in a binary form, with positive or negative classification based on posterior probability threshold of 0.5. Results are presented in reference to random patient selection.

Condition	Context example	Objective	Strategy	Action	Starting/fixed prior	Results in training set (positive misclassified among cleared)	Results in missing-data set (positive misclassified among cleared)	
Testing shortage	Testing capacity is limited to a third of candidates only	Maximize number of infected patients tested	Prioritize TP identification	Fine-tune prior until positive reach testing capacity	0.5	130% increase in actual infected patients tested (prior=0.3482)	100% increase in actual infected patients tested (prior=0.9706)	
Lack of essential workforce	Professionals with high risk of nocosomial or work-related transmission	Keep symptomatic, non-infected essential workers in duty	Search for evident non-infected workers (TN identification)	All workers are considered as infected, unless model says otherwise	0.9999	19.4% of total workforce cleared (0%)	52% of total workforce cleared (6.25%)	
Lack of essential workforce	Professionals with medium to low risk of transmission	Keep symptomatic, non-infected essential workers in duty	Find ideal balance to simultaneously, maximize both TN and TP	Use intersection of sensitivity and specificity curves from training set	0.2933	69.0% of total workforce cleared (5%)	81.5% of total workforce cleared (6.6%)	
Limited medical access	Medical assistance limited to 20% of symptomatic individuals only	Avoid contagion exposure of non-infected patients in ER during medical assistance	Eliminate non-infected from candidates for medical assistance (TN identification)	Fine-tune prior to select most likely negative results. Select remanining set for medical assistance	0.5	35.6% decrease in non-infected patients exposure (prior=0.0954)	18.8% decrease in non-infected patients exposure (prior=0.4888)	
Note:

TP, True Positive; TN, True Negative.

High accuracy in qRT-PCR result prediction is achieved based on hemogram information only. Further analysis performed on the original data (not shown) suggest that additional clinical results can improve prediction efficiency. This conclusion is in accordance with previous findings suggesting biochemical and immunological abnormalities, in addition to hematologic alterations, can be caused by COVID-19 disease (Henry et al., 2020). In this context, the relevance of data employed to generate ML models is emphasized. The use of large and comprehensive datasets, containing as much information as possible regarding clinical and laboratory findings, symptoms, disease evolution, and other relevant aspects, is crucial in devising useful and adequate models. The development of nationwide or regional databases based on local data is essential, in order to capture epidemiological idiosyncrasies associated with such populations (Terpos et al., 2020). Also, natural differences in hemogram results from distinct demographic groups (as seen in reference values according to age, sex, or other physiological factors) can aggregate noise to the model, which can be reduced when large database are employed in model construction, and results can be devised for each ethnographic strata.

Despite having high overall accuracy, performance metrics obtained with proposed model show unequal ability to predict positive or negative results. This situation is caused by a significant imbalance in number of samples belonging to each of this qRT-PCR result groups in original data. The use of balanced data in machine learning model design is important to assure high prediction quality (Krawczyk, 2016). The option of maintaining original data in model construction was adopted, since it better represents actual COVID-19 prevalence among symptomatic patients, and therefore seems to represent a more realistic situation. Additional simulations applying a balanced model (data not shown) using positive group oversampling (to compensate its insufficiency in original data) have devised alternative models with superior predictive power. Alternative balanced model results are presented in Supplemental Material (Fig. S1). Therefore, additional positive samples will be added to the data and used in future model versions.

As a perspective, collection of hemogram results from asymptomatic patients (in addition to symptomatic individuals) can be used to evaluate the utility of this approach on the detection of asymptomatic infections, in order to provide alternatives in diagnostics, especially in a context of testing deficiency. A web-based application was developed by the authors, in which hemogram data can be introduced for a single individual, along with prior probability of infection, based on data used to generate the present model. The online tool is available at http://sbcb.inf.ufrgs.br/covid. Future implementation will allow the upload of multiple patients simultaneously, and construction or testing of user data-derived models. This service will allow easy access and practical application of the proposed model.

Conclusion

Machine learning models based on hemogram data can be employed in COVID-19 pandemics management, in order to assist strategical medical decisions in different scarcity scenarios. The proposed Naïve-Bayes model has the flexibility to be applied in a large variety of possible critical conditions, and can be adjusted to improve classification accuracy for a particular target group. Even though the method proposed in this work is not suitable to be used as a diagnostic technique, it can be employed to provide additional, useful information regarding data-driven resource allocation, in shortage conditions.

Supplemental Information

Supplemental Information 1 Complete dataset used in this study.

Original data was provided by a third party (Albert Einstein Hospital, São Paulo, Brazil) and it is available online at https://www.kaggle.com/einsteindata4u/covid19.

Click here for additional data file.

Supplemental Information 2 Performance metrics of alternative balanced Naïve-Bayes model.

In this case, random oversampling of positive results was employed, until sample number in each class is identical. Prior probabilities are presented in reference to positive qRT-PCR prediction. Confusion matrices (left to right) are presented for 0.9999, 0.2237 and 0.0001 prior probabilities, respectively. Sensitivity=True Positive Ratio; Sensitivity=True Negative Ratio. Random seed was set to 0 for replication purposes.

Click here for additional data file.

Additional Information and Declarations

Competing Interests

Author Contributions

Data Availability

The authors declare that they have no competing interests.

Eduardo Avila conceived and designed the experiments, performed the experiments, analyzed the data, prepared figures and/or tables, authored or reviewed drafts of the paper, and approved the final draft.

Alessandro Kahmann conceived and designed the experiments, performed the experiments, analyzed the data, prepared figures and/or tables, authored or reviewed drafts of the paper, and approved the final draft.

Clarice Alho analyzed the data, authored or reviewed drafts of the paper, and approved the final draft.

Marcio Dorn conceived and designed the experiments, performed the experiments, analyzed the data, prepared figures and/or tables, authored or reviewed drafts of the paper, and approved the final draft.

The following information was supplied regarding data availability:

The dataset used in this study was provided by a third party (Albert Einstein Hospital, São Paulo, Brazil). It is also available as a Supplemental File and at Kaggle: https://www.kaggle.com/einsteindata4u/covid19.

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
