# Peer review of "Hemogram data as a tool for decision-making in COVID-19 management: applications to resource scarcity scenarios"

_PeerJ, doi:10.7717/peerj.9482_

## Round 0.1 · original submission · Major Revisions

Your manuscript has been reviewed and requires several modifications prior to making a decision. The comments of the reviewers are included at the bottom of this letter. Reviewers indicated that the methods sections should be improved. The manuscript also needs extensive English editing because there are several typos and grammatical errors. I agree with the evaluation and I would, therefore, request for the manuscript to be revised accordingly.

Reviewer 1 ·

Basic reporting

The work presenting by the authors are interesting and addressed an important issue in the resource-limited country. However, some improvement should be implemented before it can be accepted.
1. The paper is not well written. for example, the abstract is poorly written, and is not informative. many important information is missing and readers cannot have a general idea of the study design (such as sample size) and main results (such as performance metrics of the ML model). I strongly suggest the authors to report this paper as per the TRIPOD checklist (https://med.stanford.edu/content/dam/sm/s-spire/documents/ManuscriptQualityChecklists/Tripod-Checklist-Prediction-Model-Development-and-Validation.pdf)!
2. The baseline characteristics should be reported.
3. Many other important variables such as demographics, past history of pulmonary disease and smoking; these are readily available in resource-limited country; why not include these variables for your ML model?

Experimental design

Due to the imbalanced dataset, the accuracy cannot be used for evaluating the model. since you can predict all patients without COVID-19 and the accuracy can be 90% in your situation. Thus, I suggest to report the AUROC or PR curve for the evaluation. Furthermore, You also need to compare your model to the baseline model. The baseline model refers to a naive model that predict all patients without COVID-19.

Validity of the findings

The model testing is limited by the small sample size; there are only 10 events.

Additional comments

The work presenting by the authors are interesting and addressed an important issue in the resource-limited country. However, some improvement should be implemented before it can be accepted.

·

Basic reporting

The overall quality of the writing is fine, but there are a number of grammatical and typographical mistakes that should be corrected. Please proofread.

Experimental design

The article meets the criteria, but would have been a lot stronger if a few things had been corrected:

1. The authors argue that one of the main benefits of using naive Bayes is the interpretability of the model, then state that in this case the model should not be interpreted. I agree with neither of these: the interpretability of the model is not an important benefit, and the model can be interpreted, with caution.
2. The authors completely ignore the capacity of the model to deal with missing data, when this is a key aspect of the problem
3. It is not clear whether the test set is actually resulting in fair evaluation. In the description, it is stated that it is used for training, and that the missing values are filled with the mean value _of the class_. If that is true, the test set is not unbiased and the corresponding results are not valid.
3. The graphs are not completely clear and seem to be slightly incorrect: the y-axis is not fully labelled, and results that should be 1 (the maximum possible value in this case) don't seem to be.

Validity of the findings

See above: the validity of some results is in doubt, and some results are not reported unambiguously.

The conclusions and the potential use cases are well developed.

---

## Round 0.2 · accepted · Accept

The authors addressed the reviewer's concerns and substantially improved the content of MS. So, based on my own assessment as an editor, no further revisions are required and the MS can be accepted in its current form.

Reviewer 1 ·

Basic reporting

My previous comments have been well addressed.

Experimental design

the experimental design is not solid and sound

Validity of the findings

good

Additional comments

I have no further comments.